# Remarkable Stability Improvement of ZnO TFT with Al_2_O_3_ Gate Insulator by Yttrium Passivation with Spray Pyrolysis

**DOI:** 10.3390/nano10050976

**Published:** 2020-05-19

**Authors:** Jewel Kumer Saha, Ravindra Naik Bukke, Narendra Naik Mude, Jin Jang

**Affiliations:** 1Advanced Display Research Center (ADRC), Department of Information Display, Kyung Hee University, 26, Kyungheedae-ro, Dongdaemun-gu, Seoul 02447, Korea; jewel@tft.khu.ac.kr (J.K.S.); bukke@tft.khu.ac.kr (R.N.B.); nari@tft.khu.ac.kr (N.N.M.); 2Department of Physics, Jagannath University, Dhaka 1100, Bangladesh

**Keywords:** yttrium oxide (YO_x_), ZnO thin film, aluminum oxide (Al_2_O_3_), thin film transistor, spray pyrolysis

## Abstract

We report the impact of yttrium oxide (YO_x_) passivation on the zinc oxide (ZnO) thin film transistor (TFT) based on Al_2_O_3_ gate insulator (GI). The YO_x_ and ZnO films are both deposited by spray pyrolysis at 400 and 350 °C, respectively. The YO_x_ passivated ZnO TFT exhibits high device performance of field effect mobility (μ_FE_) of 35.36 cm^2^/Vs, threshold voltage (V_TH_) of 0.49 V and subthreshold swing (SS) of 128.4 mV/dec. The ZnO TFT also exhibits excellent device stabilities, such as negligible threshold voltage shift (∆V_TH_) of 0.15 V under positive bias temperature stress and zero hysteresis voltage (V_H_) of ~0 V. YO_x_ protects the channel layer from moisture absorption. On the other hand, the unpassivated ZnO TFT with Al_2_O_3_ GI showed inferior bias stability with a high SS when compared to the passivated one. It is found by XPS that Y diffuses into the GI interface, which can reduce the interfacial defects and eliminate the hysteresis of the transfer curve. The improvement of the stability is mainly due to the diffusion of Y into ZnO as well as the ZnO/Al_2_O_3_ interface.

## 1. Introduction

The demand for low voltage and high-performance thin-film transistors (TFTs) for next-generation displays encourages the research towards oxide semiconductor. Metal oxides are favorable semiconductor for substituting amorphous Si and low temperature poly-Si (LTPS) for display applications, particularly when high transparency is required in visible range [1,2,3]. Among various metal oxides, zinc oxide (ZnO) has attracted much interest due to its wide range of bandgap (~3.23 eV), tunability of optical and electrical properties, and nontoxic nature [4]. Pure ZnO TFT present mobility that is more competitive, as there are many studies that have reported up to 85 cm^2^V^−1^s^−1^ [5]. Several techniques are used to deposit ZnO thin film such as sputtering, chemical vapor deposition, pulsed laser deposition, spin coating, and spray pyrolysis [6,7,8,9,10,11,12]. Among these techniques, spray pyrolysis offers vacuum-free, large area deposition with low-cost [13]. The method involves spraying a solution containing soluble constituent atoms of the desired compound on to a heated substrate. The apparatus, which is needed to carry out the chemical spray process, consists of an atomizer the spray solution and a substrate heater. The sprayed droplets reaching the substrate undergo pyrolysis decomposition and form a single crystallite or a cluster of crystallites of the products as a film. In this technique, the chemicals vaporized and react on the substrate surface after reaching on it. To improve the quality of the film, some of deposition parameters, such as flow rate, distance between substrate holders (a hot plate is widely used) and the nozzle, hot plate temperature, and frequency of the movement of nozzle can be controlled. Deposition by spray-pyrolysis gives a dense ZnO film when comparing with inkjet printing and spin coating [13].

Even though high electrical performances of Nano-crystalline ZnO TFTs have been achieved, reliability and stability are the main challenges for display application. The hysteresis of the solution processed ZnO TFT on Al_2_O_3_ (high-k dielectric) is due to the oxygen related defects at the interface [14,15,16]. When the TFTs are exposed to air, desorption/adsorption of water molecule and O_2_ can affect the TFT performance by increasing the hysteresis behavior with time [17]. To protect the metal-oxide (M-O) TFTs from environment, several types of passivation layers, such as Al_2_O_3_, Y_2_O_3,_ SiO_2_, and SiNx, have been studied [18,19]. The bond dissociation energy of Y-O (714.1 ± 10.2 kJ/mol) is higher than Zn-O (159 kJ/mol), which indicates that more M-O-M network is formed and defects are reduced by Y doping [19,20,21]. Additionally, for long-term stability, Y_2_O_3_ passivation can be a good candidate because of its high oxygen bond dissociation energy. By Y_2_O_3_ passivation, the bias stability can be improved by reducing the deep level defects in ZnO and the desorption of oxygen from the back channel can be suppressed [20]. Recently, our group reported the solution-processed Y_2_O_3_ passivation layer on oxide TFTs [19,20]. It was found that the Y atoms could diffuse into the oxide semiconductor and reduce oxygen related defects.

In this work, we study the effect of yttrium oxide (Y_2_O_3_) passivation layer on the performance of ZnO TFT while using Al_2_O_3_ as a gate insulator. The Y_2_O_3_ and ZnO layers were deposited by spray pyrolysis at the substrate temperature of 400 and 350 °C, respectively. The depth profiles of atoms for the Y_2_O_3_/ZnO/Al_2_O_3_ layers have been studied by X-Ray Spectroscopy (XPS) to understand the interactions between the layers. Inter-diffusion of Yttrium (Y) into bulk ZnO and ZnO/Al_2_O_3_ interface can reduce the oxygen related defects. This improves the bias stability of ZnO TFT.

## 2. Materials and Methods

### 2.1. Material Preparation

Zinc acetate dihydrate (Zn(CH_3_COO)_2_ ·2H_2_O, product from Mumbai, India) and ammonium acetate (CH_3_CO_2_NH_4,_ product from Tokyo, Japan) precursors from Sigma Aldrich (99.999%) were dissolved into the solvent of 2-methoxyethanol (CH_3_OCH_2_CH_2_OH, product from Lyon, France) in order to synthesize 0.1 M zinc oxide (ZnO) solution. A 0.2 M aluminum oxide (Al_2_O_3_) precursor solution was made by dissolving aluminum chloride (AlCl_3,_) (Sigma Aldrich, 99.999%, product from St. Louis, MO, USA) into a mixed solvent of acetonitrile and ethylene glycol. Consequently, 0.1 M of yttrium oxide (YO_x_) precursor solution was prepared by dissolving Yttrium (III) nitrate hexa-hydrate (Y(NO_3_)_3_ · 6H_2_O, product from St. Louis, MO, USA) in 2-methoxyethanol. All of the precursor solutions were prepared under an N_2_ environment and then stirred at least 2 h to obtain a transparent solution. A 0.45 μm polytetrafluoroethylene (PTFE) filter was used to get the desired precursor solutions.

### 2.2. Device Fabrication

We fabricate the bottom-gate, bottom-contact (BG-BC) ZnO TFT while using Al_2_O_3_ as a GI. Figure 1a shows the cross-sectional view of BG-BC ZnO TFT and Figure 1b shows the optical image of the TFT after fabrication with the channel width and length of 50 μm and 10 μm, respectively. A 40 nm thick molybdenum (Mo) layer was deposited on a glass substrate by DC sputtering. The Mo film was patterned and etched for the gate electrode by photolithography. Subsequently, Aluminum oxide (Al_2_O_3_) thin film was deposited on the patterned Mo backplane by spin coating at 2000 rpm for 30 s. Afterwards, the film was cured on hot plate (250 °C) for 5 min. and exposed under UV lamp (A low-pressure mercury lamp was used as light source having wavelengths of 185 & 254 nm.) at 100 °C for UV/O_3_ treatment for 5 min. The process was repeated twice to obtain the thickness of ~40 nm, and then annealed in a furnace at 350 °C for two hours. Subsequently, the Al_2_O_3_ film was patterned by photolithography and wet etched for via-holes. A 100 nm thick Indium-zinc-oxide (IZO) layer was deposited by direct current (DC) sputtering and then patterned for the source/drain (S/D) electrodes while using conventional photolithography. A 25 nm of ZnO semiconductor was deposited by spray pyrolysis at the substrate temperature of 350 °C [12]. The movement of nozzle was controlled horizontally and vertically for n uniform film. The distance between the spray nozzle and the substrate maintained about 8 cm. The flow rate of precursor solution was 3 mL/min. and each spray cycle over 15 cm × 15 cm substrate is 60 s. The film was deposited at 350 °C and the process was repeated five times to get the 25 nm thick ZnO film. The ZnO layer was patterned by photolithography and wet etched for the active island. The channel width and length of TFTs are 50 and 10 μm, respectively. Finally, the YO_x_ film with thickness of 35 nm was deposited at 400 °C by spray pyrolysis as a passivation layer. YO_x_ film was deposited using the same flow rate of the precursor solution and then patterned using photolithography.

### 2.3. Characterization Techniques

We characterized the ZnO and YOx films by the scanning electron microscopy (SEM), X-ray diffraction (XRD), X-ray photoelectron spectroscopy (XPS), and scanning transmission electron microscope (STEM). The electrical properties of TFTs were measured using Agilent 4156C semiconductor parameter analyzer (Hewlett-Packard, Seoul, South Korea). The transfer curves were measured with the drain voltage (V_DS_) of 0.1 V and the gate voltage (V_GS_) was swept from −5 to +5 V. In the saturation region (V_DS_ ≥ V_GS_ − V_TH_), the mobility was calculated using the equation I_DS_ = (W/2L) μ_sat_ C_i_ (V_GS_ − V_TH_)^2^, where I_DS_, V_GS_, μ_sat_, W, L, V_TH_, and C_i_ are the drain current, gate voltage, saturation mobility, channel width, channel length, threshold voltage, and gate insulator capacitance, respectively. The V_TH_ was determined from the intercept of x-axis from √(I_DS_) versus V_GS_ plot by linear extrapolation and the saturation mobility (μ_sat_) from the slope of the linear part of the curve. The subthreshold swing (SS) was obtained from the equation SS = dV_GS_/d(log I_DS_) over the current range of 10 pA ≤ I_DS_ ≤ 100 pA with V_DS_ = 0.1 V.

## 3. Results and Discussions

Figure 1c shows the scanning transmission electron microscopy (STEM) image of the Y_2_O_3_/ZnO/Al_2_O_3_/Mo layers in a TFT. The image reveals uniform and continuous Y_2_O_3_/ZnO/ Al_2_O_3_/Mo interface. The thicknesses of ZnO and Y_2_O_3_ are ~25 and ~35 nm, respectively. Figure 1d,e show the fast Fourier transform (FFT) patterns for the areas indicated by the yellow circle for Y_2_O_3_ and green circle for ZnO, which reveal the nano-crystalline structure of ZnO and amorphous structure of Y_2_O_3,_, respectively. In addition, we confirmed the nanocrystalline structure of ZnO and amorphous structure of Y_2_O_3_ by XRD as shown in Appendix A. The XRD spectrum shows three strong peaks corresponding to (100), (002), and (101) planes, with the most preferred orientation being (002) plane of the hexagonal wurtzite structure. On the other hand, the Y_2_O_3_ film has no XRD peak because of amorphous phase. Appendix A shows the SEM of the ZnO and Y_2_O_3_ surface, where we can see the ZnO film are nano-crystalline with grain size of 30 nm. The ZnO film shows dense and uniform distribution of grains, which are advantageous to the electron transport. The surface morphologies of Al_2_O_3_ and ZnO thin films were studied by atomic force microscopy (AFM) measurement (Scale: 5 μm × 5 μm). Appendix A shows the surface morphology of Al_2_O_3_ thin film with root means square (RMS) roughness of 0.16 nm, whereas Appendix A shows the R_rms_ of ZnO of 1.28 nm. Therefore, the smooth surfaces of both ZnO and Al_2_O_3_ thin films are good for carrier transport [22]. In both cases of films, such as dense and uniform distribution of grains, including smooth surface, leads to lower density of defects (charge traps and recombination centers).

X-ray photoelectron spectroscopy (XPS) analysis reveals the chemical binding states of the elements and their distributions in the film. The energy of an X-ray used in XPS with particular wavelength (beam of monochromatic Al Kα X-rays, E_photon_ = 1486.7 eV). Figure 2a,b show wide-scan spectra (on the surface) of ZnO film and Y_2_O_3_ passivation layers. At the surface of the ZnO film, the peaks of Zn2p and O1s confirm ZnO formation. The presence of Y3d and O1s at the surface of YO_x_ film confirms the formation of Y_2_O_3_. The Y3d binding energy peak shows two splitting orbitals Y3d_5/2_ and Y3d_3/2_ positioned at 156.5 and 158.7 eV, respectively, as shown in Figure 2c. This confirms the formation of Y_2_O_3_ [21]. The atomic percentages of C, O, Zn, Al, and Y at the surface of ZnO without Y_2_O_3_ are 18, 42, 40, 0, and 0%, respectively. Whereas, the atomic percentages are 12.45, 63, 0, 0, and 24.55% at the surface of Y_2_O_3_, respectively [20].

The presence of carbon usually acts as carrier recombination centers. The percentage of C high only at the surface not in the bulk. Carbon is the material, which always present in atmosphere, so nano level layer is deposited on the samples. In general, the properties of carrier transport for ZnO semiconductor are closely related to the chemical states of ZnO thin film, such as a metal-oxygen bond (M–O), oxygen vacancies (Vo), and hydroxyl groups (–OH). XPS depth analysis was carried out for ZnO/Al_2_O_3_ without and with Y_2_O_3_ passivation to explore the chemical states of ZnO thin film and its interface with the GI. From the depth profile, the formation of individual layers of Y_2_O_3,_ ZnO, and Al_2_O_3_ are confirmed, as shown in Figure 2d. The presence of Y at the interface of Y_2_O_3_/ZnO, in bulk ZnO and the interface of ZnO/Al_2_O_3_ confirm the diffusion of Y to ZnO until GI. This passivates the defects at the interface [19,20,21].

Through the Gaussian–Lorentzian fitting method, the O 1s intensity peak is deconvoluted into three peaks at 530 eV, 531 eV, and 532 eV, corresponding to M–O, V_o_ and M–OH, respectively. The O1s deconvolution on the surface of ZnO is shown in Figure 3a, where the M–O bonding is 66.30% and the defects (oxygen vacancy and –OH group) are 33.70% (23.53% and 10.17%) for the unpassivated ZnO. Whereas, the passivated ZnO shows the increase of M-O bonding and reduction of defects on Y_2_O_3_/ZnO, as shown in Figure 3b. M–O bonding and defects at the interface of Y_2_O_3_/ZnO are 69.86% and 30.14%, respectively. The O1s deconvolution at the ZnO/Al_2_O_3_ interface without and with Y_2_O_3_ passivation are shown, respectively, in Figure 3c,d. The M–O bonds increase from 70.23 to 72.95% and the defects (oxygen vacancy and –OH group) decrease from 29.77 to 27.05% by Y_2_O_3_ passivation. The improvements in the ZnO and ZnO/Al_2_O_3_ interface are due to the diffusion of Y at the bulk of ZnO as well as at the ZnO/Al_2_O_3_ interface [20,21]. Even a slight change in film composition by higher bond dissociation energy, such as Y–O (714.1 ± 10.2 kJ/mol), can enhance the electrical properties [20,21].

Figure 4a,b show the transfer characteristics of ZnO TFT without and with Y_2_O_3_ passivation, respectively. The transfer curve with hysteresis characteristics and gate leakage currents are shown as a function of V_GS_. The electrical properties of unpassivated ZnO TFT are the μ_FE_ of 42.66 cm^2^V^−1^s^−1^, V_TH_ of 0.58 V, and SS of 172.40 mV/dec, whereas passivated ZnO TFT exhibits the μ_FE_ of 35.36 cm^2^V^−1^s^−1^, V_TH_ of 0.49 V, and SS of 128.40 mV/dec. The anti-clockwise hysteresis for the unpassivated ZnO TFT implies for the electron detrapping from high-k gate dielectrics at the interface between gate insulator and active layer, which generate extra carriers in the channel at negative gate bias [23]. After Y_2_O_3_ passivation, the defects at the ZnO/Al_2_O_3_ interface has reduced due to Y diffusion, which reduces the trapping and detrapping of carriers and provides a negligible hysteresis. The output curves of ZnO TFT without and with passivation are shown, respectively, in Figure 4c,d. The output curves are showing clear pinch-off and saturation behavior. There are no current crowding in the linear region, indicating the excellent ohmic contact between active layer and S/D. The reduction of mobility with zero hysteresis voltage is most probably due to the excess amount of Y (around 10%, as highlighted by XPS depth analysis), which can suppress the carrier concentration, thus reducing mobility [24]. The interface state density N_SS_, as calculated from SS for unpassivated and passivated ZnO TFTs, are 2.69 × 10^12^ and 7.38 × 10^11^ cm^−2^eV^−1^, respectively. Table 1 shows the summary of device performances, such as μ_FE_, SS, and I_ON_/I_OFF_, including the data from the literatures [19,20,21,25,26,27,28,29].

Among all reports, our spray pyrolyzed Y_2_O_3_ passivated ZnO TFT shows superior mobility. The higher mobility with good stability is due to the interface quality between gate insulator and semiconductor layer, which has further confirmed from the TEM image and XPS analysis. We measured 15 TFTs with same W/L (50 μm/10 μm) at different locations to check the uniformity of spray coated TFTs at different position over 7.5 cm × 7.5 cm substrate glass. The histograms show the comparison of μ_FE_, V_TH_, and SS values for ZnO TFTs without and with Y_2_O_3_ passivation, as shown in Figure 5a–c. The average μ_FE_ for unpassivated and passivated ZnO TFTs are 43.13 ± 3.48 and 36.88 ± 3.75 cm^2^/V.s, respectively. The average V_TH_ for unpassivated and passivated ZnO TFTs are 0.61 ± 0.05 and 0.50 ± 0.04 V, respectively. In addition, the average SS for unpassivated and passivated ZnO TFTs are 170.40 ± 9.22 and 126.40 ± 10.98 mV/dec, respectively. Table 2 summarizes electrical performances of ZnO TFT fabricated by spray pyrolysis without and with Y_2_O_3_ passivation and statistical data for 15 ZnO TFTs.

For the electrical stability of the spray coated ZnO TFTs without and with Y_2_O_3_ passivation, the positive bias stress (PBS) at V_GS_ of +5 V and negative bias stress (NBS) at V_GS_ of −5 V were applied by varying the stress time from 0 to 1 h. Figure 6a,b show the evolution of transfer curves under PBS and NBS were measured by sweeping gate voltage from −5 to +5 V at the same drain voltage (V_DS_) of 0.1V for passivated ZnO TFT.

Additionally, Appendix A show the evolution of transfer curves under PBS and NBS for unpassivated ZnO TFT. The ∆V_TH_ for unpassivated ZnO TFT under PBS and NBS are 0.3 and 0.25 V, respectively, whereas the Y_2_O_3_ passivated ZnO TFT shows the ∆V_TH_ values of 0.03 and 0.05 V, respectively. The positive shift of the transfer curve is understood by the trapping of electrons at the ZnO/Al_2_O_3_ interface [30,31]. Small ∆V_TH_ indicates fewer defects at the interface of the GI and the active layer. The negligible change in the SS after bias stress from 0 to 1 h indicates very small trapping sites at the interface [32,33].

The positive-bias-temperature-stress (PBTS) (at V_GS_ = 5 V for 1 h) for the ZnO TFT was investigated. Figure 6c shows the evolution of transfer curves of the ZnO TFT under PBTS for 1 h in dark at 60 °C, as measured by sweeping V_GS_ from −5 to +5 V at a V_DS_ of 0.1 V. The positive ∆V_TH_ (0.78 V) is due to the electron trapping at the ZnO/Al_2_O_3_ interface for unpassivated ZnO TFT, as shown in Appendix A. The Y_2_O_3_ passivated ZnO TFT is highly stable under PBTS for 1 h in dark showing ∆V_TH_ of 0.15 V [34,35,36,37]. Therefore, the improvement in the operational stability is due to the significant reduction of ZnO/Al_2_O_3_ interface trap by interfacial defects. Furthermore, we also checked the long-term stability of Y_2_O_3_ passivated ZnO TFT in ambient air for six months and measured the TFT performance after a certain interval. There is no significant change observed in the performance of ZnO TFT with Y_2_O_3_ passivation shown in Appendix A. Therefore, the spray coated Y_2_O_3_ passivation layer at 400 °C is quite good to protect the device from the absorption or desorption of oxygen present in air.

## 4. Conclusions

We studied the impact of yttrium oxide (Y_2_O_3_) passivation layer on the performance of zinc oxide (ZnO) thin film transistor (TFT) based on Al_2_O_3_ gate insulator (GI). Y_2_O_3_ and ZnO films are both deposited by spray pyrolysis at 400 and 350 °C, respectively. Y_2_O_3_ passivated ZnO TFT exhibits field effect mobility (μ_FE_) of 35.36 cm^2^/Vs, threshold voltage (V_TH_) of 0.49 V, subthreshold swing (SS) of 128.4 mV/dec., and hysteresis voltage (V_H_) of ~0 V. The negligible threshold voltage shift (∆V_TH_) of 0.15 V under positive bias temperature stress (PBTS) was found in the passivated device. The yttrium (Y) can improve the device performance due to its higher bond dissociation energy with oxygen. It can also protect the channel from moisture absorption. In addition, the diffusion of Y until the ZnO/Al_2_O_3_ interface reduces the interfacial defects. The Y diffusion, as confirmed from the XPS analysis, leads to the decrease of oxygen related defects and to increasing the M-O-M bond network at the ZnO/Al_2_O_3_ interface. Therefore, the solution-processed Y_2_O_3_ passivation on metal oxide TFTs is a good approach for next-generation display technology.

## Figures and Tables

**Figure 1 nanomaterials-10-00976-f001:**
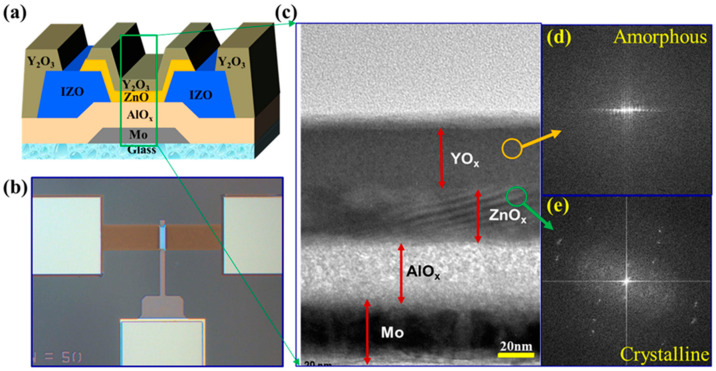
(**a**) Schematic diagram and (**b**) optical image of the bottom gate bottom contact ZnO TFT with Y_2_O_3_ passivation by spray pyrolysis. (**c**) Cross-sectional transmission electron microscope (TEM) image of a ZnO TFT. Fast Fourier Transform (FFT) of (**d**) yellow circled area confirms the amorphous nature of Y_2_O_3_ layer deposited at 400 °C and (**e**) green circled area confirms the hexagonal structured ZnO deposited at 350 °C by spray pyrolysis.

**Figure 2 nanomaterials-10-00976-f002:**
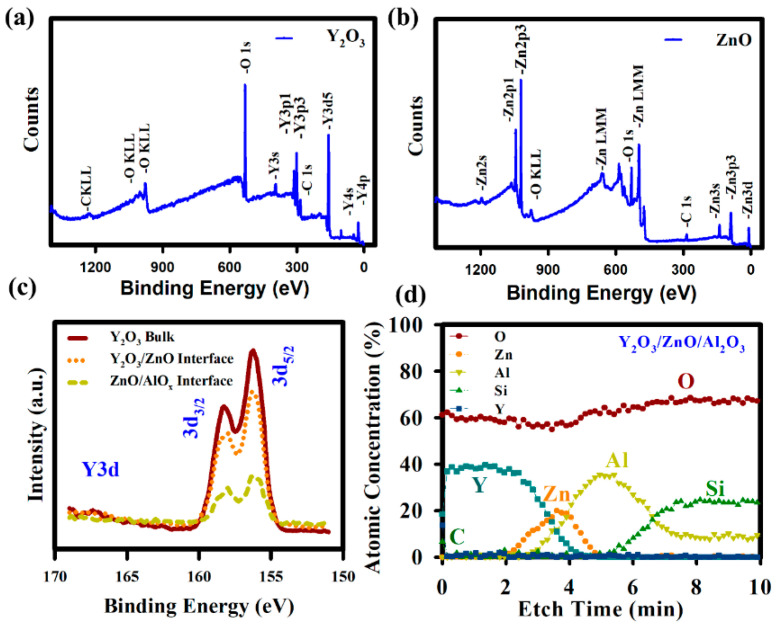
X-ray photoelectron spectroscopy (XPS) survey spectra of (**a**) Y_2_O_3_ and (**b**) ZnO films. (**c**) The Y3d peaks at different depths of Y_2_O_3_/ZnO/Al_2_O_3_ film. Y could be found at the bottom interface (ZnO/Al_2_O_3_). (**d**) XPS depth profiles of O, Y, C, Zn, Al, and Si of Y_2_O_3_/ZnO/Al_2_O_3_ film.

**Figure 3 nanomaterials-10-00976-f003:**
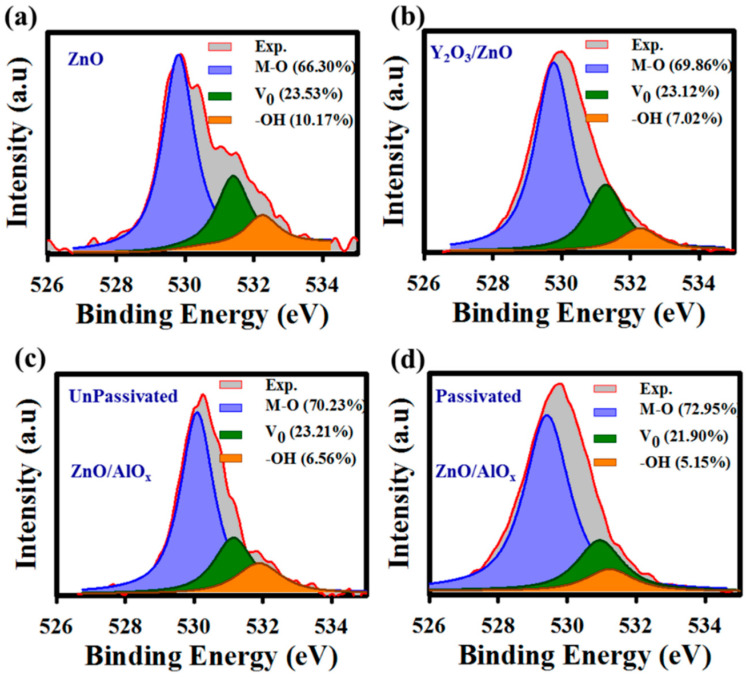
XPS spectra analysis of the deconvoluted O 1s peak (**a**) in the bulk of ZnO and (**b**) at the interface of Y_2_O_3_/ZnO. Deconvoluted O 1s peak at the interface of ZnO/Al_2_O_3_ (**c**) without and (**d**) with Y_2_O_3_ passivation. The individual contribution of metal oxide (M–O), oxygen vacancy (Vo) and metal hydroxyl group (M–OH) are shown by area under the blue line (~529.5 eV), green line (~531 eV), and orange line (~532 eV), respectively. After Y_2_O_3_ passivation the Vo and –OH concentrations are reduced at the interface of Y_2_O_3_/ZnO as well as interface of ZnO/Al_2_O_3_.

**Figure 4 nanomaterials-10-00976-f004:**
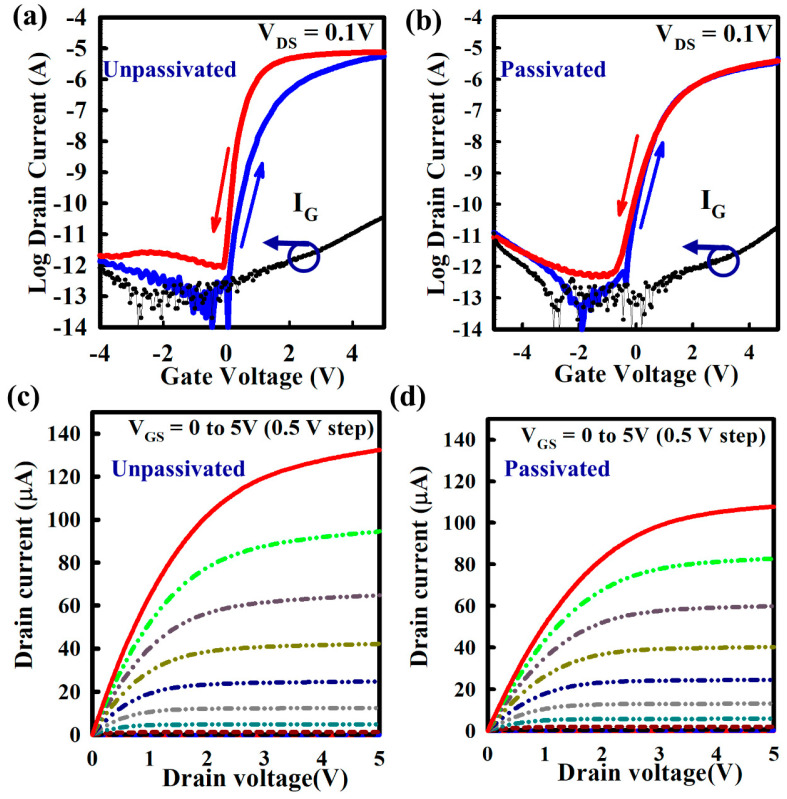
Electrical performance of ZnO TFT without and with Y_2_O_3_ passivation by spray pyrolysis. (**a**,**b**) Plots of the transfer characteristics with hysteresis and gate leakage currents (I_G_) as a function of V_GS_ for (**a**) without and (**b**) with Y_2_O_3_ passivation. (**c**,**d**) Output curves of ZnO (**c**) without and (**d**) with Y_2_O_3_ passivation. The hysteresis voltage of the thin film transistor (TFT) was taken at I_DS_ = 10^−10^ A with forward sweep from −5 to +5 V and reverse sweep from +5 to −5 V at the same drain voltage (V_DS_ = 0.1 V). Output curves were measured by varying V_GS_ from 0 to +5 V with a 0.5 V step.

**Figure 5 nanomaterials-10-00976-f005:**
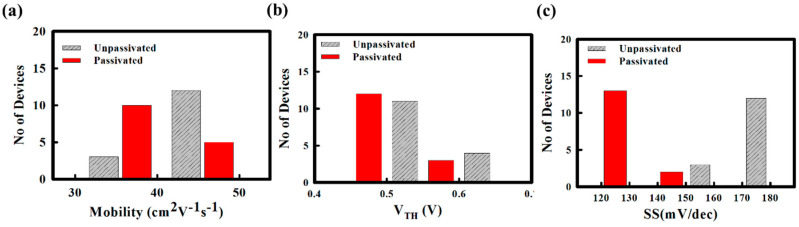
Statistical analysis of the electrical performances of the ZnO TFTs without and with Y_2_O_3_ passivation. The histogram of 15 ZnO TFTs for (**a**) mobility, (**b**) V_TH_, and (**c**) subthreshold swing (SS), respectively.

**Figure 6 nanomaterials-10-00976-f006:**
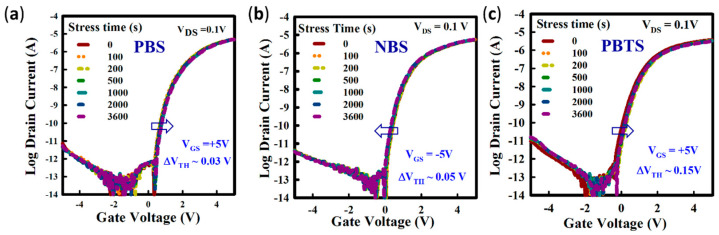
The evolution of transfer curve under bias stress for the ZnO TFTs with Y_2_O_3_ passivation. Transfer curves of ZnO TFTs under (**a**) positive bias stress (PBS), (**b**) negative bias stress (NBS), and (**c**) positive-bias-temperature-stress (PBTS) at 60 °C. Thetransfer curve of the TFT measured by sweeping V_GS_ from −5 to +5 V at the drain voltage, V_DS_ = 0.1 V for 1 h.

**Table 1 nanomaterials-10-00976-t001:** The summary of processing methods and electrical performances (field effect mobility, subthreshold swing, and current on/off ratio) of the solution processed oxide TFTs with various passivation layers.

Active/GI	Passivation Layer	Processing Method	Substrate Temp. (°C)	μ_FE_ [cm^2^V^−1^s^−1^]	SS [mV/dec]	I_on_/I_off_	Year
InOx/SiO_2_	AlOx	SC	400	0.02	730	10^6^	2014 ^(25)^
ZnO/SiO_x_	PDMS	SC	150	0.50	240	10^6^	2012 ^(26)^
IGZO/SiO_2_	Y_2_O_3_	SC	250	21.31	160	10^8^	2014 ^(19)^
IZTO/ZrO_x_	Y_2_O_3_	SC	350	4.75	114	10^9^	2016 ^(20)^
IGZO/SiO_2_	Y_2_O_3_	SC	350	11.10	140	10^8^	2012 ^(27)^
GdInOx/AlO_x_	Y_2_O_3_	SC	350	9.47	79	10^7^	2017 ^(21)^
ZnO/SiO_2_	PCBA	SC	150	4.50	-	10^6^	2018 ^(28)^
IGZO/SiO_2_	HfO_x_	SC	250	9.60	350	10^8^	2017 ^(29)^
IGZO/SiO_2_	Y_2_O_3_	SC	250	11.30	267	10^3^	2017 ^(29)^
IGZO/SiO_2_	PMMA	SC	250	9.51	320	10^8^	2017 ^(29)^
ZnO/AlO_x_	Y_2_O_3_	SP	350	35.36	128	10^8^	This work
**GI: Gate Insulator, SC: Spin Coating, SP: Spray Coating, Temp.: temperature**							

**Table 2 nanomaterials-10-00976-t002:** Summary of the electrical performances of 15 ZnO TFTs fabricated by spray pyrolysis without and with Y_2_O_3_ passivation.

ZnO TFT	µ_FE_ (cm^2^V^−1^ s^−1^)	V_th_ (V)	SS (mV/dec)
Without passivation	43.13 ± 3.48	0.61 ± 0.05	170.40 ± 9.22
With passivation	36.88 ± 3.75	0.50 ± 0.04	126.40 ± 10.98

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
