# Peer review of "Remarkable Stability Improvement of ZnO TFT with Al2O3 Gate Insulator by Yttrium Passivation with Spray Pyrolysis"

_nanomaterials, 2020, doi:10.3390/nano10050976_

Round 1

Reviewer 1 Report

In this work, Saha et al. report on the performance of a ZnO TFT with Al2O3 gate insulator where a Y2O3  passivation layer is used for the first time. Yttria showed to improve the performance of TFT in terms of electrical properties, as well as of stabillity. Moreover, authors make a comprehensive and detailed XPS analysis, and provide a physical interpretation of the results. The main element of novelty is the use of the spray coating technique for passivation, which shows some benefits if compared to spin coating usually employed with other TFTs.

As a whole, it’s a good work, and deserves publication in Nanomaterials journal.

However, there are some major issues that must be mandatorily addressed before publication, in my opinion:

1) The first and most crucial aspect to clarify is the mobility of charge carriers, on which it seems that authors made a little bit confusion. They report an average mobility value for passivated and non-passivated TFTs of 35.36 and 42.66 cm2V/s, respectively. Therefore, according to these values, mobility is worse in the passivated case, and this sounds very strange to me. Indeed, if the diffusion of Y at the interface at the ZnO/Al2O3 has proven to reduce interface defects, this should imply an improvement of the charge carrier mobility, and not a decrease. Authors state that “The reduction of mobility with zero hysteresis voltage suggests that the interface defects has reduced due to the Y diffusion into ZnO/ Al2O3 interface [24]”. However, it’s the opposite in my opinion, because the reduction of interface defects always improves mobility, as reported, by the way, in Ref.[24] cited by the authors themselves. Another demonstration of the confusion made by the authors is given in Fig. 5(a), where it’s clearly highlighted that the average carrier mobility in passivated TFTs is BETTER than in the non-passivated case! I invite authors to carefully check the mobility values (which have been swapped by mistake, in my opinion), and to reconsider their conclusions on the effect of interface defects on the charge carrier transport.

2) In section 2.2, several details on the device fabrication and on the transistor size are missing. Have the patterning of all the structures (gate electrode, via-holes, IZO layer, etc.) been made by standard optical lithography? Or e-beam lithography? What are the thicknesses of Al2O3, IZO, ZnO, and Y2O3 layers? Why has a UV/O3 treatment been performed for Al2O3deposition? What are the channel length and width (they should appear here, and not after in the text)? Why have depositions of Al2O3 and Y2O3 layers  been performed by subsequent steps until the “desired thickness” is obtained, and not by a unique step? Finally, I also suggest to move the description of the device (first lines of section 3) in the device fabrication section. Before describing the fabrication procedure, it’s indeed appropriate to show and describe the device structure reported in Fig. 1(a).

3) The methodology used for stress tests should be better explained. Authors state that “For the electrical stability of the spray coated ZnO TFTs, the positive bias stress (PBS) (at VGS = 5 V for 1 h) and negative bias stress (NBS) were investigated. The evolution of transfer curves under PBS or NBS measured by sweeping gate voltage from -5 to + 5 V and +5 to -5V at the same drain voltage (VDS = 0.1V at VGS of +5 V) are shown in Fig. 6(a and b)”. First, the negative bias voltage -5V in the NBS case should be clearly indicated in the text, as made for the positive case. Also, it seems to me from Figure 6 that authors have used different stress times (not only 1h), varying from 100 s to 3600 s (1 h). So why in the text only a stress time equal to 1 h is mentioned? I invite authors to clarify and explain the methodology used, so to avoid misunderstandings. Moreover, PBS and NBS results should be also shown for the non-passivated case, in order to compare them to the passivated cases of Fig. 6a and 6b. This has been done for the PBTS case (Fig. 6c and Fig. S4), why not also for PBS and NBS?

Minor comments:

  • Check mobility value in the abstract and the conclusion; I think it’s 42.66 cm2V/s and not 35.36 cm2V/s).
  • Remove PBTS acronym from the abstract (it’s not used further in the abstract .
  • In the introduction, the bandgap value of ZnO should be indicated.
  • In the introduction, few lines describing the spray pyrolysis technique should be added, otherwise it’s not useful to list all the deposition parameters (flow rate, substrate-holder distance, hot-plate temperature, nozzle speed, etc,.)
  • In “High-k dielectric”, “k” is lowercase (not “High-K”).
  • In the introduction, define the acronym “M-O” (even if it’s quite intuitive that is “Metal-Oxide”).
  • Change the title of section 2.3. Replace “Physical characteristics” with “Characterization techniques”.
  • At the end of section 2.3 (last line) replace ID with IDS.
  • When stating that “dense and uniform distributions of grains are advantageous to the electron transport”, as well as that “smooth surfaces of both ZnO and Al2O3 thin films are for good for carrier transport” authors should also explain why. In both cases, there is indeed a lower density of defects (charge traps and recombination centers) which could significantly affect carrier mobility. It should be clearly stated.
  • X-ray energy used for XPS should be specified.
  • Atomic percentage of carbon is significant (it varies from 12.45 to 18%). Authors state that it’s due to contamination during XPS. But the percentage is quite high to be only ascribed to contamination. How can they exclude that C was not incorporated during film deposition? Could authors elaborate a little bit more on this?
  • Replace the term “suppressing sites” with “recombination centers”.
  • The statement “The Y3d binding energy peak shows two splitting orbitals Y3d5/2 and Y3d3/2 positioned at 156.5 and 158.7 eV, respectively as shown in Fig. 2d” should be moved before in the text, soon after “The presence of Y3d and O1s at the surface of YOx film confirms the formation of Y2O3.” Therefore, positions of Fig. 2c and 2d should be swapped (Fig. 2d will be the new Fig. 2c and viceversa).
  • The statement “The bond dissociation energy of Y-O (714.1 ± 10.2 kJ/mol) is higher than Zn-O (159 kJ/mol), which indicates more M-O-M network is formed and defects are reduced by Y doping [19-20,22]” at the end of page 4 should be moved to the introduction section, and merged with the statement “For long-term stability, Y2O3 passivation can be a good candidate because of its high oxygen bond dissociation energy (Y−O = 714.1 ± 10.2 kJmol-1).”
  • Replace “Lorenz” with “Lorentzian”.
  • The increase of M-O bonds and the decrease of defects by Y diffusion are about 3.6% and 2.7%, respectively. How do they authors explain, if possible, that such low percentages are anyway sufficient to significantly improve the TFT performance and stability? I suggest to add a few lines about this consideration.
  • When writing “excellent ohmic contact between active and S/D”, add “layer” to “active”.
  • Few lines should be added commenting the results shown in Table 1. More specifically, it should be highlighted that results of the authors’ work are the best in term of mobility.
  • Table 2. As I pointed out in the general comments, check the mobility values for non-passivated and passivated TFTs, and compare them with those reported in Figure 5a. There must be something wrong.
  • Figures 4a and b. The Y-axis reporting gate leakage current values is missing. Also, the arrow of the gate leakage current plot should point right, not left (where drain current log values are reported).
  • In the caption of Fig. 6, replace “60C” with “60°C”.
  • In the statement “The positive VTH (0.78 V) is due to the electron trapping at the ZnO/ Al2O3 interface as shown in Fig. S4” it should be specified that it refers to a non-passivated TFT.

Author Response

Dear professor

Thank you for your valuable comments. I tried to answer all of your comments.

Thanks

Jewel Kumer Saha

Reviewer 2 Report

In their paper, the Authors report about the fabrication and the analysis of a thin-film transistor based on a ZnO channel, with IZO drain and source electrodes, Mo bottom gate, Al2O3 gate insulator and Y2O3 passivation layer, with ZnO and Y2O3 deposited by spray pyrolysis.
Through microscopy, spectroscopy, and electrical characterization, the Authors show that the Y2O3 passivation layer has improved the behavior of the transistor, in terms of reduction of defects and higher stability.

The article is very interesting and well written; therefore, I surely advice its publication.

I suggest only the correction of some typos, that I list in the following.

First of all: at page 5, under Figure 3, change "with and without Y2O3 passivation, respectively" with "without and with Y2O3 passivation, respectively".

Then:
1) page 1, line 7 of the Introduction: change "though" with something more proper (and same thing for "thoughtful" at line 3 of the Introduction)
2) page 1, line 16 of the Introduction: "challanges" -> "challenges"
3) page 1, line 17 of the Introduction: "histerisis" -> "histeresis"
4) page 2, line 1: "histerisis bahaviour" -> "histeresis behaviour"
5) page 2, line 11: "respctively" -> "respectively"
6) page 2, line 34: "substarte" -> "substrate"
7) page 2, line 41: "by measuring" -> "by"
8) page 3, two lines before Figure 1: "are for good" -> "are good"
9) page 4, line 3 of the caption of Figure 2: "ta" -> "at"

Author Response

(The authors gave the same response as above.)

Reviewer 3 Report

The authors report on the use of yttrium oxide as a passivation layer for thin film transistors comprised of zinc oxide films with an aluminum oxide gate dielectric. The authors fabricate the devices using spray pyrolysis, a cost effective method for large scale production. They characterize the devices using scanning electron microscopy, transmission electron microscopy, x-ray photoelectron spectroscopy, atomic force microscopy, and electrical measurements. They conclude that yttrium diffuses through the zinc oxide layer and into the interface between the aluminum oxide and zinc oxide. This reduces interface defects that lead to hysteresis and improves overall device stability.

I think the authors have presented the results clearly and they have drawn sound conclusions from their results. I would recommend publication of the manuscript provided the authors consider the following comments.

- The authors have performed a bias stress test for the devices and the passivation layer does a good job of improving device stability. I wonder if the authors have also considered long term stability? How do the transfer curves stand against longevity? Do you see any change with time?

- It seems to me that if you have removed interface defects or traps that the mobility should increase in the passivated devices but the opposite is true. The mobility is seen to decrease in the passivated devices. Can the authors discuss this in the text in a few lines to make this point clear?

Author Response

(The authors gave the same response as above.)

Round 2

Reviewer 1 Report

I thank the authors for having welcomed my comments and accepted my suggestions. 

Also, the question about mobility has finally been clarified. Authors provided a new reference (Appl. Surf. Sci. 475, pp. 565-570, 2019) where it's demonstrated that Y can reduce or increase mobility depending on the atomic concentration. In the authors' case, Y atomic concentration is around 10%, and according to the new reference it causes a mobility reduction, as confirmed by the authors' results.

There is only one thing left before publication.

In the revised version of the paper, authors still state that “The reduction of mobility with zero hysteresis voltage suggests that the interface defects has reduced”. However, I think that this statement could be misleading, because in most cases (in the field of semiconductor-based FETs), interface defects cause a mobility reduction. So it may sound strange to the reader. According to the convincing explanation given by the authors in the response letter, the statement should be replaced by the following: “The reduction of mobility with zero hysteresis voltage is most probably due to the excess amount of Y (around 10%, as highlighted by XPS depth analysis), which can suppress the carrier concentration, thus reducing mobility [24]”.

Once addressed this minor issue, the paper is ready for publication, in my opinion.

Author Response

Dear Professor

Thank you for your valuable suggestions.

We changed the statement according to your statement.

Sincerely 

Jewel Kumer Saha
